# STYLEMASTER: TOWARDS FLEXIBLE STYLIZED IMAGE GENERATION WITH DIFFUSION MODELS

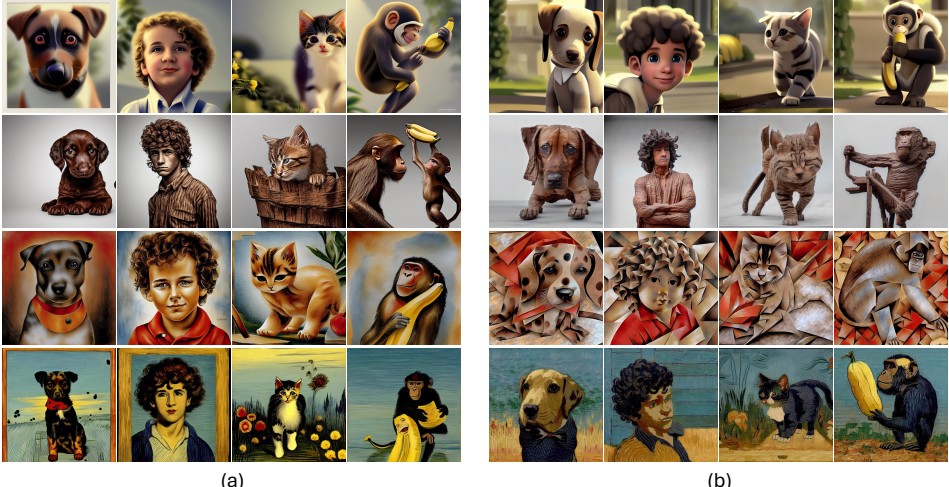

Figure 1: (a) Results generated by StyleAdapter. (b) Results generated by our proposed StyleMaster, which can better stylized the images based on the reference images. Compared with StyleAdapter, our method can generate more consistent and delicate stylized images with different styles. For simplicity we only present the generated images here. Styles from up to bottom: *pixar, wooden, cubism, Van Goah*. For the style reference images please refer to the supplementary material.

## ABSTRACT

Stylized Text-to-Image Generation (STIG) aims to generate images based on text prompts and style reference images. We in this paper propose a novel framework dubbed StyleMaster for this task by leveraging pretrained Stable Diffusion (SD), which addresses previous problems such as misinterpreted style and inconsistent semantics. The enhancement lies in two novel modules: multi-source style embedder and dynamic attention adapter. In order to provide SD with better style embeddings, we propose the multi-source style embedder, which considers both global and local level visual information along with textual information, thereby offering both complementary style-related and semantic-related knowledge. Additionally, aiming for better balance between the adapter capacity and semantic control, the proposed dynamic attention adapter is applied to the diffusion UNet in which adaptation weights are dynamically calculated based on the style embeddings. Two objective functions are introduced to optimize the model alongside the denoising loss, which can further enhance semantic and style consistency. Extensive experiments demonstrate the superiority of StyleMaster over existing methods, rendering images with variable target styles while successfully maintaining the semantic information from the text prompts.

## 1 INTRODUCTION

Stylized Image Generation (SIG), which generates images with specific styles, has broad academic value and practical applications in fields like art and film. With the rise of diffusion-based generative

models Ho et al. (2020); Rombach et al. (2022), research has shifted from traditional style transfer to Stylized Text-to-Image Generation (STIG). In this task, one or several style reference images are given as condition, and various images are generated based on these style conditions along with additional information such as text prompts. Generally, STIG approaches provide enhanced flexibility despite their complex, mixed input conditions, making them highly relevant for real-world applications.

Recently, StyleAdapter Wang et al. (2023b), built on pretrained Stable Diffusion (SD), has emerged as a representative and effective STIG method. It processes reference images through a style embedding module, injecting these embeddings into the diffusion UNet via cross-attention modules to guide style incorporation during denoising and generate stylized images. The framework has also been successfully extended in recent work Wang et al. (2024). However, despite its effectiveness with specific styles and prompts, StyleAdapter faces key challenges: (1) **misinterpreted style**, where generated images fail to fully capture complex reference styles, and (2) **inconsistent semantics**, where elements from reference images leak into the output, misaligning with text prompts. These issues arise from the design of StyleAdapter, particularly its use of CLIP Radford et al. (2021), which focuses on local patch-level patterns while neglecting critical global patterns. Furthermore, by injecting style embeddings into all cross-attention layers of the diffusion UNet, the model biases text-prompt integration, leading to semantic inconsistencies.

In this paper, we propose StyleMaster, a novel STIG framework featuring *advanced extraction* and *injection of style information*. Our method builds on the basic pipeline of StyleAdapter but introduces specific enhancements for style embedding extraction and injection.

**For style extraction**, we propose a multi-source structure to address the limitations of CLIP's patch-level information. Specifically, in addition to using CLIP-based patch embeddings, we incorporate global-level VGG descriptors, which are integrated into the attention process via adaptive scaling and shifting. Moreover, we utilize the semantic knowledge from reference image captions. To prevent this abstract semantic content from influencing the style embedding, we introduce a negative embedding branch that removes semantic information. By combining these different latent spaces, our method captures a more comprehensive representation of target styles while avoiding semantic leakage. **For style embedding injection**, we advocate that limiting the attachment of adapters only to specific parts of the UNet, such as the upsampling layers, could prevent semantic distortion. However, this direct limitation reduces the capacity of adapters and worsens the issue of misinterpreted style. To resolve this, we propose a dynamic attention adapter that generates dynamic weights from the style embeddings. These weights adapt both self-attention and cross-attention layers in the diffusion UNet, allowing for more precise and flexible style adaptation. This ensures that the generated images maintain both the intended style and semantic consistency with the text prompts.

Furthermore, we enhance the model with objectives beyond the standard noise prediction loss commonly used in diffusion models. We introduce a Gram consistency loss, augmenting the reference images with two sets of transformations: one set preserves the original style, while the other adopts a distorted style. We then compute Gram matrices of the estimated denoised results and these transformed reference images as their style-aware statistics. By applying a triplet loss among these matrices, the model is encouraged to generate images with more robust and consistent styles when processing different reference images. Additionally, we utilize a semantic disentanglement loss to mitigate the inconsistent semantics problem by contrasting the style embeddings against reference text embeddings, while ensuring they remain similar to the reference image embeddings.

To show the effectiveness of our proposed method, we conduct extensive experiments among various styles containing both one-shot and multi-shot settings. We show that our method can significantly outperform baseline methods including StyleAdapter, generating correct styles and avoiding inconsistent semantics. In summary, the contributions of this work are as follows:

1) *Enhanced Framework for STIG*: We present StyleMaster, a novel framework for Stylized Text-to-Image Generation (STIG) that enhances the existing StyleAdapter pipeline. StyleMaster significantly improves style embedding extraction and injection, effectively addressing issues such as misinterpreted styles and inconsistent semantics.

2) *Multi-Source Structure for Style Extraction*: Our StyleMaster introduce a multi-source approach that integrates CLIP-based patch embeddings, global-level VGG descriptors, and semantic knowledge from image captions, capturing comprehensive target style representations while preventing semantic leakage through a negative embedding branch.

3) *Dynamic Attention Adapter for Style Injection*: To enhance style embedding injection, StyleMaster introduces a dynamic attention adapter that generates weights from style embeddings, enabling precise adaptation of self-attention and cross-attention layers in the diffusion UNet. This ensures that generated images maintain the intended style and semantic consistency with text prompts.

4) *Enhanced Training Objectives*: our model incorporates objectives beyond standard noise prediction loss, including a novel Gram consistency loss that promotes robust and consistent styles through triplet loss on Gram matrices from transformed reference images. Additionally, a semantic disentanglement loss contrasts style embeddings with reference text embeddings while maintaining similarity to reference image embeddings, addressing inconsistent semantics.

## 2  RELATED WORK

**Text-to-image diffusion models.** Diffusion models have been proven to be a powerful family of generative models. DDPM Ho et al. (2020) originated to propose the framework by modeling the mapping between Gaussian distribution and image distribution with the forward diffusion and inverse denoising process. Based on that Latent Diffusion Model (LDM) Rombach et al. (2022) largely improved the practical usage by leveraging diffusion model to latent space instead of pixel space, which leads to commonly-known text-to-image diffusion models such as Stable Diffusion (SD), Midjourney and DALLE-3 Betker et al. (2023). Other works focus on improve the diffusion model structure. For example, DiT Peebles & Xie (2023), MDT Gao et al. (2023) and PIXART-$\alpha$ Chen et al. (2023) utilize the transformer instead of UNet structure, which can be better scaled to larger model size. Blattmann et al. (2022) and Zhang et al. (2023b) leverage ideas of Retrieval Augmented Generation (RAG) to generate images based on other retrived images which provide extra knowledge. Yang et al. (2024) propose to leverage the LLMs for planning the text-to-image problems. Our work is built on pretrained SD models, in which the attention mechanism merges conditional information from text prompts to images. Different from the previous works, we focus on designing extra attention adaptation so that the knowledge contained in the style reference images can be smoothly embedded into the denoising process, leading to stylized images.

**Stylized image generation.** Among all conditional image generation tasks, stylized image generation has long been a highlighted one. Most previous works focus on style transfer, *i.e.*, transfer the style of a content image given another style refernce image. For example, Gatys et al. (2016) solved this problem by optimizing the style-related statistics. Li et al. (2017) equipped this method with explanability. MicroAST Wang et al. (2023a) proposed to speed up such framework by abandoning the complex visual encoder and utilizing a dual-modulation strategy. Yang et al. (2023) leverages diffusion models, in which the style-aware guidance is used to generate the wanted style. InST Zhang et al. (2023c) realized style transfer by inverting the content image to noise and then re-generate it with the condition control of style images. Apart from these style transfer methods, StyleAdapter Wang et al. (2023b) proposes a new framework which can generate images directly from style reference images and text prompts without content images. Our work mainly follows StyleAdapter to present a generalized stylization method. Different from StyleAdapter, we analyze the role of style reference images and text prompts in the generation process. Based on that, we propose a novel module to extract more representative style embeddings, which are then injected into noise space with our proposed dynamic adapter.

## 3  PRELIMINARY: STABLE DIFFUSION

Diffusion models model the data distribution $p_\theta(\mathbf{x}_0)$ of clean data $\mathbf{x}_0$ by progressively denoising a standard Gaussian distribution, of which the learning process is instantiated as denoising score matching. Stable Diffusion (SD) extends such a model to text-to-image based on text prompt $p$. With pre-trained VQ-VAE Van Den Oord et al. (2017) containing encoder $\mathcal{E}$ and decoder $\mathcal{D}$, SD allows the model to focus more on the semantic information of data and improves efficiency. A diffusion UNet is used to predict the noise, in which attention mechanism is adopted. Specifically, for the $l$-th layer, self-attention is first used to interact among spatial features: $z^l = Attention(W_Q^l \cdot z^l, W_K^l \cdot z^l, W_V^l \cdot z^l)$, where $Attention$ denotes the attention operator, $z^l$ denotes latent embeddings of the $l$-th layer, $W_Q, W_K, W_V$ denotes the projection layers of self-attention. After that the cross-attention is utilized to merge condition information such as text prompt: $\hat{z}^l = Attention(\hat{W}_{Q_t}^l \cdot$

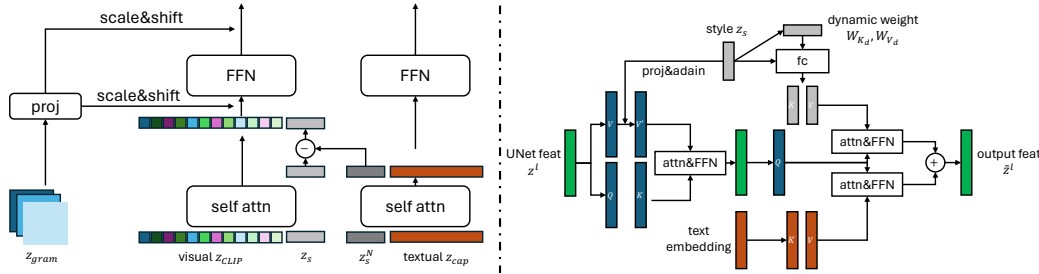

Figure 2: Overview of our proposed StyleMaster. Concretely, the **multi-source style embedder** (Sec. 4.1) aggregates style patterns contained in each reference image simultaneously from global and local levels. Style embeddings are then engaged in the denoising procedure through the proposed **dynamic attention adaptation** (Sec. 4.2), which guides both the attentions in diffusion UNet to properly merge style and semantic information from different sources. During training augmented input images are used as style reference images, through which objectives as described in Sec. 4.3 are used to optimize the model. During inference the style reference images are achieves with manual assignment instead of using augmentation of a specific image.

Figure 3: Left: multi-source style embedder. Right: dynamic attention adapter.

$z^l, \hat{W}_{K_t}^l \cdot z_{text}, \hat{W}_{V_t}^l \cdot z_{text})$, where $z_{text}$ denotes text prompt embedding, $\hat{W}_Q, \hat{W}_K, \hat{W}_V$ denotes the projection layers of cross-attention. The training objective of SD is as follows:

$$\mathcal{L}_{noise} = \mathbb{E}_{\mathcal{E}(x), \epsilon \sim \mathcal{N}(0,1), t} \left[ \left\| \epsilon - \epsilon_\theta \left( z^t, t \right) \right\|_2^2 \right], \qquad (1)$$

where $t$ is uniformly sampled from $\{0, ..., T\}$, $z^t$ denotes noisy latent at $t$-th timestep.

## 4 METHODOLOGY

We focus on reference-based Stylized Text-to-Image Generation (STIG) in this paper. Formally, a reference style image set $\mathcal{I}_s = \{\mathbf{I}_{style}^i\}_{i=1}^{N_s}$, where $N_s$ denotes the number of reference images, together with text prompt $p$ are given as condition information. $N_s$ can be variable among different trials to describe different style concepts. The model is required to generate image $\mathbf{I}$ that shares the same style pattern with $\mathcal{I}_s$ and same semantic meaning with text prompt $p$. To solve this task we present a novel framework named StyleMaster based on SD, as shown in Fig. 2, which will be introduced in this section.

### 4.1 MULTI-SOURCE STYLE EMBEDDER

Given the successful application of text-to-image based on SD, an ideal reference-based stylization should necessarily depend on extracting style embeddings that are as representative as the text embeddings resulted from strong embedders such as CLIP and T5, from reference images.

StyleAdapter Wang et al. (2023b) adopts CLIP visual encoder Radford et al. (2021) to extract the patch-level features from style reference images, which are then processed with cross-attentions using learnable style tokens. However, StyleAdapter still suffers from issues such as misinterpreted style and inconsistent semantics, *i.e.*, the semantic knowledge in reference images is generated during the denoising procedure. The former problem can be attributed to the fact that CLIP-based patch embeddings concentrate more on local patterns. In this way, the style embeddings fail to hold the global style patterns, leading to misinterpreted styles. Meanwhile, since CLIP image embeddings are aligned with their captions, they inevitably contain information about content in the images. Incorporating such information in style embeddings can lead to inconsistent semantics. Moreover, since SD models different kinds of information across different denoising timesteps, providing identical conditional information means SD has to extract the required information on its own, thus leading to heavier learning burden for SD.

To solve these problems, we propose leveraging knowledge from multiple sources to complement the style embedding. Specifically, given a style image set $\mathcal{I}_s$, we extract three different kinds of features with pretrained models: (1) **local style-aware features**: Following StyleAdapter, we utilize CLIP to encode each style image $\mathbf{I}_{style}^i$ into latent patch tokens $\tilde{z}_{CLIP}^i \in \mathbb{R}^{c \times (wh)}$, where $c, h, w$ denote latent channel, height and width of CLIP features. Then we apply discrete wavelet transform (DWT) to $\tilde{z}_{CLIP}^i$, leading to low-frequency features $\tilde{z}_{CLIP,lf}^i$ and high-frequency counterparts $\tilde{z}_{CLIP,hf}^i$. All low and high frequency features from $\mathcal{I}_s$ are then concatenated together along the token dimension, resulting in $z_{CLIP}$. (2) **global style-aware features**: Inspired by previous style transfer methods Gatys et al. (2016), we adopt the Gram matrix to represent global-level style information. Specifically, $\mathbf{I}_{style}^i$ is processed with VGG-19 Simonyan & Zisserman (2014) to obtain the *relu3_1* feature $z_{vgg}^i \in \mathbb{R}^{c' \times h'w'}$, where $c', h', w'$ denote the latent channel, height and width of VGG feature map. Then the Gram matrix can be calculated as $z_{gram}^i = z_{vgg}^i \cdot {z_{vgg}^i}^T$, which is flattened to a vector with dimension $\mathbb{R}^{c^2}$ afterwards. Then Gram matrices for different style images are averaged to get the final representation $z_{gram}$. (3) **semantic-aware features**: We adopt CLIP text encoder to extract the text embedding $z_{cap}^i$ of captions of $\mathbf{I}_{style}^i$, which are then concatenated into $z_{cap}$. During training, the captions are provided in the training set, while during inference, we adopt BLIP Li et al. (2022) to first annotate the style reference images.

Generally, each token in $z_{CLIP}$ contains local-level style features, while $z_{gram}$ describes more abstract style information. On the other hand, $z_{cap}$ contains the semantic information of $\mathcal{I}_s$, which should be eliminated in the final style embedding to avoid inconsistent semantics. To properly make use of these embeddings, we propose a novel dual-branch structure as shown in Fig. 3(a). Concretely, several learnable style tokens $z_s$ are first attached to $z_{CLIP}$, with their replication $z_s^N$ denoted as negative semantic tokens attached to $z_{cap}$:

$$\hat{z}_{CLIP} = z_{CLIP} \|_t (z_s + \delta_t), \quad \hat{z}_{caption} = z_{cap} \|_t z_s^N \tag{2}$$

where $\cdot\|_t\cdot$ denotes concatenation along the token dimension, $\delta_t$ is the same time embedding of the denoising timestep as used in diffusion UNet. $\hat{z}_{caption}$ is then individually processed with several transformer layers to aggregate the information between style tokens and text embedding. For $\hat{z}_{CLIP}$, we adopt a modified version of transformer layer. Specifically, $z_{gram}$ is first projected to scaling and shift coefficients. These coefficients are applied to the self-attention procedure in the same way as adaLN Perez et al. (2018). Before processing the attention result with FFN, the style token part in $\hat{z}_{caption}$ is subtracted from the counterpart in $\hat{z}_{CLIP}$.

Our design enjoys three major merits. First, the module is timestep-aware. Since the frequency-aware style tokens are merged with time embedding, different denoising steps can thus model different information. Second, the global information in $z_{gram}$ can guide the model to better concentrate on style-related knowledge, thus avoiding those irrelevant but repetitive local patterns shared among style images. Third, the negative semantic tokens generally contain more abstract content information rather than style. Consequently, subtracting it from $\hat{z}_{CLIP}$ can help alleviate the problem of inconsistent semantics. While some captions may describe the style of images, the model can learn to maintain this knowledge during training thanks to the training strategy described in Sec. 4.3. By using the proposed module, we can learn more representative and generalizable style embeddings which can better facilitate the stylization process described as follows.

## 4.2 DYNAMIC ATTENTION ADAPTATION

The extracted style embedding $z_s$ from $\mathcal{I}_s$ as described above can then be used to adapt the pretrained SD to guide the denoising process based on style information. Thanks to the design of self-attention and cross-attention mechanism in diffusion UNet, such adaptation can be simply instantiated as an extra cross-attention module that is parallel to the original prompt-based cross-attention, as adopted in StyleAdapter. However, we empirically find that such a method is suboptimal in a large amount of cases, leading to severe semantic inconsistency. A straightforward solution is cut down the number of extra attention modules so that only upsample layers in diffusion UNet are adapted. In this way, the text prompt can dominate the cross-attention in half of the UNet, consequently resulting in better semantics in the generated images. However, this can decrease the capacity of adapters, leading to less preferable stylization. To this end, we propose adopting a dynamic adaptation strategy which is applied to both self-attention and cross-attention (Fig. 3(b)).

**Dynamic self-attention adapter.** As discussed in previous works Hertz et al. (2023), the projected value tensor $W_V^l \cdot z^l$ in the self-attentions contributes to the texture of generated images. Therefore we introduce a dynamic self-attention adapter module based on adaIN. Formally, for the $l$-th self-attention layer, we project $z_s$ with a linear layer and adjust $W_V^l \cdot z^l$ according to statistics of $z_s$:

$$\hat{\mathbf{V}}^l = \mu(f_{proj-SA}^l(z_s)) + \frac{\sigma(f_{proj-SA}^l(z_s))}{\sigma(W_V^l \cdot z^l)} * (W_V^l \cdot z^l - \mu(W_V^l \cdot z^l)) \tag{3}$$

where $f_{proj-SA}^l$ denotes dynamic projection layer, $\mu, \sigma$ denote mean and standard deviation. By rescaling $\mathbf{V}^l$, the information contained in $z_s$ can be directly embedded into the image feature without destroying the structure and semantic meaning of the generated image.

**Dynamic cross-attention adapter.** To adapt the cross-attention layers, we follow the idea of StyleAdapter to adopt the dual-path cross-attention mechanism. Basically, for the $l$-th cross-attention layer, besides the original cross-attention performed between text embedding $z_{text}$ and image embedding $z^l$ as in Sec. 3, an extra style-aware cross-attention is added as $\tilde{z}^l = Attention(\hat{W}_{Q_t}^l \cdot z^l, \hat{W}_{K_s}^l \cdot z_s, \hat{W}_{V_s}^l \cdot z_s)$ with additional learnable parameters $\hat{W}_{K_s}^l, \hat{W}_{V_s}^l$. Then $\hat{z}^l + \lambda\tilde{z}^l$ is fed into the following feed-forward networks, where $\lambda$ is learnable coefficient.

To enhance the capacity, we further propose a dynamic cross-attention adapter. Specifically, we first project the statistics of $z_s$ to a layer specific latent space:

$$z_s^l = f_{proj-CA}^l(\mu(z_s)\|_c \sigma(z_s)) \tag{4}$$

where $f_{proj-CA}$ denotes a linear projection layer, $\cdot\|_c\cdot$ denotes concatenation along the channel dimension. Then two weight generators instantiated as linear layers are applied to $z_s^l$, resulted in two dynamic weights $W_{K_d}^l, W_{V_d}^l$ with dimension $d * d^l$. These two features are reshaped into linear layer weights and used to transform $z_s$. After that the style-aware cross-attention is modified as

$$\tilde{z}^l = Attention(\hat{W}_{Q_t}^l \cdot z^l, (\hat{W}_{K_s}^l + W_{K_d}^l) \cdot z_s, (\hat{W}_{V_s}^l + W_{V_d}^l) \cdot z_s) \tag{5}$$

In this way, the key and value projections are partially dependent on $z_s$, resulting in a more complex transformation of $z_s$ and leading to better capacity. To make this module parameter-efficient, we adopt a grouping strategy, *i.e.*, channels of $z_s$ in each group share the same dynamic weight to produce $W_{K_d}^l$ and $W_{V_d}^l$, hence lighter weight generators can be used to generate dynamic weights.

## 4.3 TRAINING OBJECTIVES

To train the model such that it can generate images that are conforms to both the style information from style reference images and semantic information from text prompts, we introduce a mixed training objectives including three terms as follows.

$$\mathcal{L} = \mathcal{L}_{noise} + \mathcal{L}_{disen} + \mathcal{L}_{style} \tag{6}$$

where $\mathcal{L}_{noise}$ is the noise prediction loss as in Eq. 1. $\mathcal{L}_{disen}$ denotes a semantic disentangle loss applied to style embedding $z_s$. To ensure that the proposed style embedding model can get rid of the semantic information contained in $\mathbf{I}_{style}$ when producing $z_s$, $\mathcal{L}_{disen}$ is designed by enlarging the similarity between $z_s$ and $z_{CLIP}$ while decreasing the similarity between $z_s$ and text embedding

$z_{text}$. Formally, $L_{disen} = sim(z_{cap}, z_s) - \delta sim(z_{CLIP}, z_s)$, where $\delta$ is hyper-parameter set as 0.1, $sim$ denotes cosine similarity. As discussed in Sec. 4.1, the text embedding represents more abstract semantic information than the image embedding. Therefore this loss term can help the model avoid the possibility of semantic leakage, thus leading to better style embedding. On the other hand, to enhance the style consistency, we propose to regulate the Gram matrix of $\hat{x}^0$, which is the noisy estimation from $z^t$ and can be calculated as

$$\hat{z}^0 = \frac{z^t - \sqrt{1 - \bar{\alpha}^t} \epsilon^t}{\sqrt{\bar{\alpha}^t}}, \quad \hat{x}^0 = \mathcal{D}(\hat{z}^0) \tag{7}$$

Specifically, we apply several rigid transformations such as random rotation and cropping to $\mathcal{I}_S$ to get a new image $\mathbf{I}_{pos}$ that has the same style as $\hat{x}_0$. Then elastic transformation and color jitter are also applied to $\mathcal{I}_S$. The resulted $\mathbf{I}_{neg}$, while sharing similar semantic object to $\mathbf{I}_{inp}$, barely inherits the style from it. Then the objective is a triplet loss which can be written as

$$\delta_p = \sum |\mathcal{G}(\phi_{vgg}(\hat{x}_0)) - \mathcal{G}(\phi_{vgg}(\mathbf{I}_{pos}))| \tag{8}$$

$$\delta_n = \sum |\mathcal{G}(\phi_{vgg}(\hat{x}_0)) - \mathcal{G}(\phi_{vgg}(\mathbf{I}_{neg}))| \tag{9}$$

$$\mathcal{L}_{style} = max\{\delta_p - \delta_n + 0.1, 0\} \tag{10}$$

where $\mathcal{G}$ denotes the Gram matrix of features. By optimizing this loss term, the model is encouraged to learn more detailed style information, thus leading to better results. In total, during training, only the style embedder and the added adapters are trained with objectives as in Eq. 6. Those used backbones such as VGG, CLIP and original parameters from SD are not trained. During inference, we directly use the trained models to generate stylized images without any test-time optimization.

## 5 EXPERIMENTS

### 5.1 IMPLEMENTATION DETAIL

**Dataset.** We follow StyleAdapter to adopt LAION-Aesthetic 6.5+ as the training set, which contains about 600k images. For each input image during training, we use its augmented variants as the style reference images. For evaluation we adopt 50 prompts used in StyleAdapter, and select 20 styles covering color, texture and global layout. More details are presented in the supplementary material.

**Experiment setting.** Our experiments cover both one-shot and multi-shot settings. To make the evaluation more challenging, the shot number, *i.e.*, number of reference images, varies from 2 to 5 among different styles in the multi-shot setting. We use all 20 styles for multi-shot experiments and 10 of them for one-shot experiments.

**Training details.** We adopt AdamW as optimizer with 1e-5 learning rate. Our model is trained for 200,000 iterations on 8 V100s with 8 batch size on each gpu, which takes about 3 days.

**Competitor.** We include extensive methods as our competitor. For 1-shot experiment, MicroAST Wang et al. (2023a), StyleAlign Hertz et al. (2023), StyTR$^2$ Deng et al. (2022), and StyleAdapter Wang et al. (2023b) are adopted. For multi-shot experiment, InST Zhang et al. (2023c), LoRA Hu et al. (2021), Textual Inversion (TI) Gal et al. (2022), StyleDrop Sohn et al. (2023) and StyleAdapter are adopted.

### 5.2 QUANTITATIVE RESULTS

**Objective quantitative results.** For quantitative evaluation we adopt CLIP to calculate the style similarity between generated images and target style reference images, and the semantic similarity between generated images and target text prompts. The results are presented in Tab. 1 and Tab. 2 for 1-shot and multi-shot respectively. Note that for SD1.5, InstantStyle receives incredibly high style similarity, which is not attributed to its strong performance. In fact, as we will show in the qualitative results, InstantStyle-SD1.5 totally leaks the content in the reference images into the generated images, thus leading to extremely poor text similarity. Moreover, MicroAST and StyTR$^2$ share similar text similarity to ours one-shot experiments. This is because we use pretrained SD to generate base images for them, thus the basic semantic meaning is contained in the image. However the gap in terms of style similarity is much more marginal. Our method outperforms the best baseline

Table 1: Quantitative results for one-shot setting. For all metrics, the larger score denotes the better model.

| Backbone | 1-shot Methods | Text Sim ↑ | Style Sim ↑ |
|---|---|---|---|
| | MicroAST Wang et al. (2023a) | 0.299 | 0.529 |
| | StyTR[2] Deng et al. (2022) | 0.298 | 0.541 |
| SD15 | StyleDrop Sohn et al. (2023) | 0.290 | 0.583 |
| | InstantStyle Wang et al. (2024) | 0.167 | **0.840** |
| | StyleAdapter Wang et al. (2023b) | 0.282 | 0.668 |
| | Ours | **0.299** | 0.708 |
| | StyleAlign Hertz et al. (2023) | 0.276 | 0.645 |
| SDXL | InstantStyle Wang et al. (2024) | 0.295 | 0.652 |
| | Ours | **0.311** | **0.696** |

Table 2: Quantitative results for multi-shot setting. For all metrics, the larger score denotes the better model.

| Backbone | multi-shot Methods | Text Sim ↑ | Style Sim ↑ |
|---|---|---|---|
| | InST Zhang et al. (2023c) | 0.196 | 0.692 |
| | LoRA Hu et al. (2021) | 0.237 | 0.665 |
| | TI Gal et al. (2022) | 0.268 | 0.678 |
| SD15 | StyleDrop Sohn et al. (2023) | 0.273 | 0.599 |
| | InstantStyle Wang et al. (2024) | 0.186 | **0.749** |
| | StyleAdapter Wang et al. (2023b) | 0.286 | 0.682 |
| | Ours | **0.291** | 0.719 |
| SDXL | InstantStyle Wang et al. (2024) | 0.291 | 0.645 |
| | Ours | **0.293** | **0.667** |

StyleAdapter by 0.04. As for SDXL, our method performs generally better than other competitors. The results for multi-shot setting are consistent with one-shot, thus showing the superiority of the proposed method.

Table 3: User Study for one-shot setting with SD1.5 as backbone. For all metrics, the larger score denotes the better model.

| Methods | Text Sim ↑ | Style Sim ↑ |
|---|---|---|
| MicroAST | 3.18 | 2.84 |
| StyTR[2] | **3.85** | 3.25 |
| StyleDrop | 3.11 | 2.91 |
| StyleAdapter | 3.16 | 3.30 |
| Ours | 3.20 | **3.80** |

Table 4: User Study for multi-shot setting with SD1.5 as backbone. For all metrics, the larger score denotes the better model.

| Methods | Text Sim ↑ | Style Sim ↑ |
|---|---|---|
| InST | 2.11 | 2.68 |
| LoRA | 2.84 | 3.34 |
| TI | 2.95 | 3.03 |
| StyleDrop | 2.71 | 3.11 |
| StyleAdapter | 3.12 | 3.78 |
| Ours | **3.14** | **3.86** |

**Subjective quantitative results.** Apart from the quantitative results in the main paper, we randomly sample 5 images for each method using SD1.5 as backbone and each style and conduct a user study. Given the abnormal performance of InstantStyle, it is not involved in this process. During the study, the participant are required to score 1-5 for each image by considering both semantic fidelity and style consistency. Then all scores are averaged. The results are shown in Tab. 3 and Tab. 4, which are generally consistent with the objective results except that the text similarity of StyTR[2] is extremely high with human scores. Since StyTR2 cannot fully stylized the base images, some of its results are exactly the same as the base images, which makes it enjoy almost the same semantic fidelity as SD, leading to better text similarity but low style similarity. On the other hand, while StyTR[2] receives good semantic fidelity, its style consistency is poor. Our method, on the contrary, can find a good balance for the trade off between semantic and style.

## 5.3 QUALITATIVE RESULTS

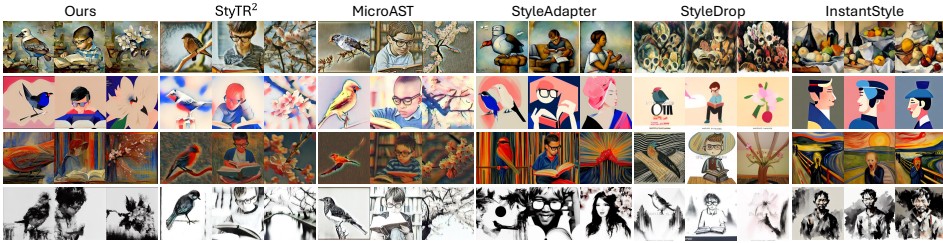

Figure 4: One-shot qualitative comparison with SD1.5 as backbone. Styles from top to bottom: *Cezanne, flat cartoon, expressionism, ink*. Prompts from left to right: *A bird in a word; A boy wearing glasses, he is reading a thick book; A cherry blossom*. For detailed reference images and comparison with SDXL as backbone, please refer to the appendix.

We present several uncurated results with SD1.5 as backbone for both settings in Fig. 4 and Fig. 5 respectively. First of all, as mentioned above, we find that InstantStyle suffers from problem of

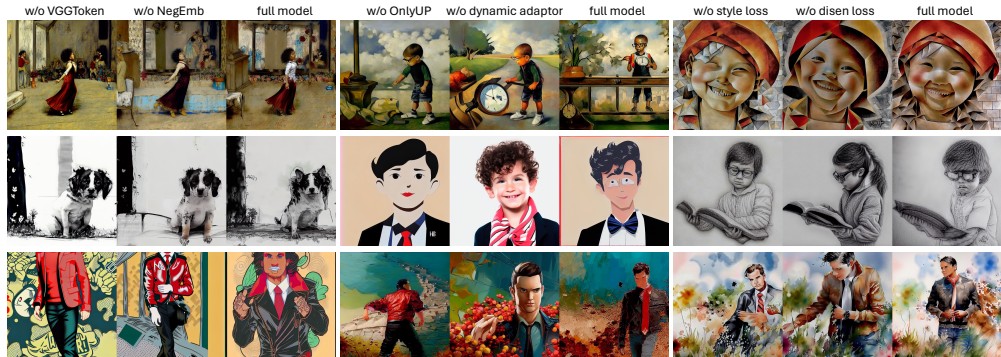

Figure 5: Multi-shot qualitative comparison with SD1.5 as backbone. Styles from top to bottom: *pencil, watercolor, Monet, impasto*. Prompts from left to right: *A robot; A modern house with a pool; A lake with calm water and reflections.* For detailed reference images and comparison with SDXL as backbone, please refer to the appendix.

Figure 6: Images generated by different model variants.

content leakage, resulting in meaningless generation. For one-shot experiment, since we use pretrained SD to first generate the content images, the style transfer based methods such as StyTR$^2$ and MicroAST can generally enjoy reasonable semantic consistency. However, such methods can only inherit the basic color information from reference images rather than the detailed style information such as shape, texture and layout, thus making them less preferable. For example, they fail to present the cubism and the curves in the second and third row. This is because these methods rely on simple representation to transfer the style-related knowledge from reference images, which leads to the problem of under-stylization. StyleAlign, on the other hand, can stylize the images better. For example the images generated by StyleAlign in the fourth row share similar style patterns with the reference image. However, we find that StyleAlign suffers from inconsistent semantics problem, which leads to inconsistent semantic meaning with the text prompt. Moreover, images generated by StyleAlign seems messy and disordered, which may be attributed to the uncontrolled shared self-attention between content and style images. The performance of StyleAdapter is much better than other competitors, while it is hard for this method to understand complex style patterns, leading to undesirable results when it comes to ink painting (fourth row). Compared with these methods, our method can learn appropriate style information from reference images, *e.g.*, the scattered color patches in the first and fourth row, and simultaneously keep the images faithful to the prompts, thus making the best of both worlds.

The multi-shot setting which is more challenging shows similar results. InST can hardly replicate the style. LoRA and TI suffer from limited style information. StyleAdapter, while utilizing a specifically-designed pipeline, shows a tendency to confuse the given styles with the photographic prior knowledge from pretrained SD. Such phenomenon is most obvious in the first row of Fig. 5, where pencil drawings are provided as reference images, but StyleAdapter generates grayscale photos. Our method, thanks to the proposed multi-source style embedder which can extract more detailed style information and the dynamic attention adaptation, can generally generate different kinds of styles with high image quality and semantic fidelity.

Table 5: Quantitative ablation study for both one-shot and multi-shot settings. For all metrics, the larger score denotes the better model.

| multi-shot Methods | One-shot | | Multi-shot | |
|---|---|---|---|---|
| | Text Sim | Style Sim | Text Sim | Style Sim |
| w/o VGGToken | 0.288 | 0.692 | 0.288 | 0.698 |
| w/o NegEmb | 0.285 | 0.689 | 0.281 | 0.696 |
| w/o OnlyUP | 0.286 | 0.697 | 0.286 | 0.702 |
| w/o dynamic adaptor | **0.301** | 0.611 | **0.299** | 0.643 |
| w/o style loss | 0.280 | 0.694 | 0.290 | 0.705 |
| w/o disen loss | 0.282 | 0.695 | 0.286 | 0.694 |
| Ours | 0.299 | **0.708** | 0.291 | **0.719** |

## 5.4 ABLATION STUDY

To further verify the efficacy of our contributions, we conduct several ablation studies on multi-shot setting. The quantitative results are shown in Tab. 5. More qualitative ablation studies are provided in the appendix.

**Design of style embedding module.** We consider two variants together with the full model for the style embedding module: not using the VGG Gram matrix to regulate the attention layers (w/o VGGToken), and not engaging the reference image captions in the extraction (w/o NegEmb). The results are shown in the left three columns of Fig. 6. The style of images generated by model without VGGToken is generally less mimic. Meanwhile, the model without NegEmb not only has worse style (first and second row) but also suffers from mistaken semantic meaning (third row).

**Design of attention adapter.** We illustrate the role of different parts in the proposed dynamic attention adapter in the middle three columns of Fig. 6. When the model adopts attention adapter for all UNet attention layers instead of only the upsample ones, the images generally have problem of mistaken semantic meaning. For example, the clock is missing in the first row, and the cloth color is mistaken in the third row. Also, it is obvious that when only using the same cross-attention adapter as in StyleAdapter, the generated images show inconsistent and undesirable styles, which can be attributed to the limited capacity to such strategy. Interestingly, we find in Tab. 5 that when not using dynamic adapter, the model has a very different tendency compared with the full model, with much better text prompt similarity but much worse style similarity. This is reasonable since the proposed dynamic adapter greatly strengthens the impact of style embedding during both self-attention and cross-attention. In this way, the cross-attention with text prompts is weakened is disguise. In general, adopting dynamic adapter can make a good balance between text prompt fidelity and style fidelity.

**Effectiveness of different objective functions.** In the right three columns of Fig. 6 we inspect the efficacy of two objectives introduced in Sec. 4.3. The results directly support our claim that the gram consistency loss can enhance the style in generated images and the semantic disentangle loss can make the model tell apart semantic and style information from reference images, thus better handling contents in the text prompts.

## 6 CONCLUSION

We try to solve Stylized Text-to-Image Generation in this paper. A novel model is proposed to solve the problems of misinterpreted style and inconsistent semantics which are suffered by previous methods such as StyleAdapter. The improvement mainly comes from the multi-source style embedder, in which multiple sources and used to achieve comprehensive style embeddings and eliminate the semantic information from style reference images, and the dynamic attention adapter, in which style embeddings dynamically interact with attention layers in diffusion UNet. Extensive experiments have been conducted to show the efficacy of our proposed method as a powerful stylization method, which can be widely applied to real-life scenarios.

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

## A  EXPERIMENT DETAILS

**Competitors.**  We follow the original training protocol to train StyleAdapter, and utilize 500 and 1000 iterations for LoRA and TI respectively to achieve suitable performance and avoid overfitting. To adapt the style transfer methods to our setting, we first utilize pretrained SD v1.5 to generate an image according to the text prompt, then transfer its style using the corresponding methods. For StyleAlign, we first adopt DDIM-inversion Mokady et al. (2023) to invert reference images back to noise, and then attach it to other images to be generated.

**Style reference images.**  To make sure our experiments is extensive enough to show the generalization ability of our method, we manually design the style reference image set, which are shown in Fig. 7 and Fig. 8, with the search results provided by Google with key words sets as the notations as depicted in the captions. Our style reference images cover different style concepts such as artistic contents, shapes, colors and textures, which can better support our conclusion that the proposed AnyArt is capable for various cases.

**Styles used in figures in main paper.**  In order to make some results (Fig. 1 and Fig. 6) in main paper simple and easier to understand, we omit the style reference images and summarize the used styles here:

- Fig. 1 from first row to fifth row: *pixar, ink, wooden, cubism, Van Goah*.
- Fig. 6 first row from left to right: *Degas, Cezanne, Cubism*.
- Fig. 6 second row from left to right: *ink, flat cartoon, pencil*.
- Fig. 6 third row from left to right: *flat cartoon, impasto, watercolor*.

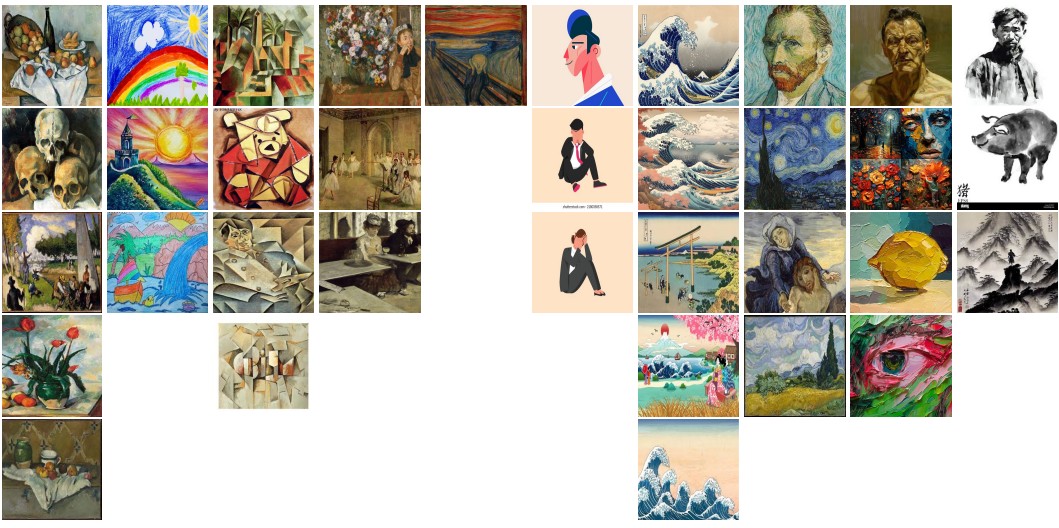

Figure 7: Style reference images used in this paper. Each column denotes a style, in which the images in the first row are used in one-shot experiments, and all images are used in multi-shot experiments. Notations for each column from left to right: *Cezanne, crayon, cubism, Degas, expressionism, flat cartoon, ukiyo, Van Goah, impasto, ink*.

**Prompts used in figures in main paper.**  All qualitative results in the qualitative ablation study of main paper use the same prompts as in the quantitative evaluation, concretely as follows,

- Fig. 6 first row from left to right: *A girl wearing a red dress, she is dancing; A little boy with glasses and a watch; A smiling little girl*.
- Fig. 6 second row from left to right: *A puppy sitting on a sofa; An curly-haired boy; A boy wearing glasses, he is reading a thick book*.
- Fig. 6 third row: *A man wearing a black leather jacket and a red tie*.

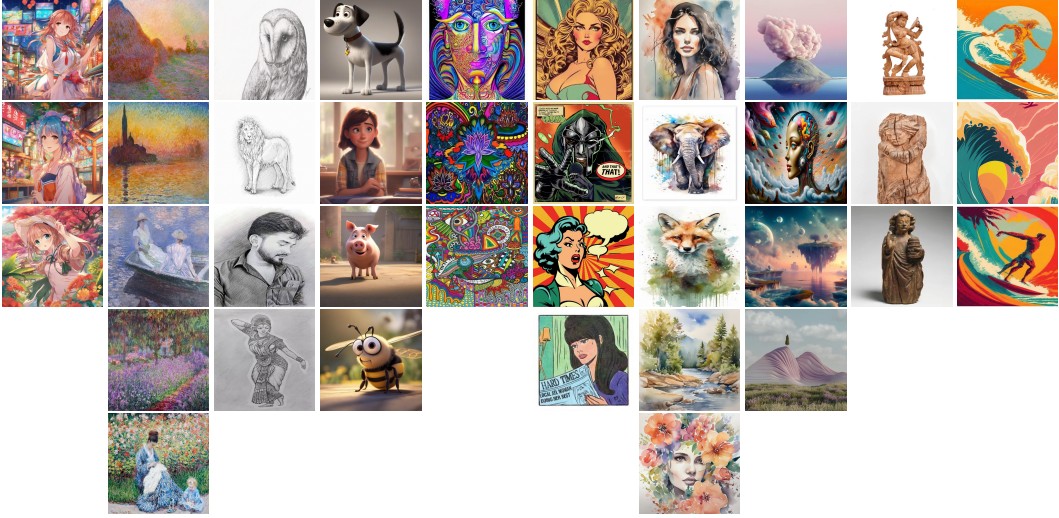

Figure 8: Style reference images used in this paper. Each column denotes a style, in which the images in the first row are used in one-shot experiments, and all images are used in multi-shot experiments. Notations for each column from left to right: *japan anime, Monet, pencil, pixar, psychedelic, america anime, watercolor, surreal, wooden, surf.*

## B    MORE DISCUSSION

**Limitations.**    We would like to highlight two limitations of our method. First, the multi-source style embedding extraction model relies on a patch-level transformer. In this way, it is less efficient for this model to process too many style reference images. While such a scenario is to some extent unrealistic, since it is generally sufficient to represent a specific style with less than 10 images, solving this problem can be related to improving vision transformer structures, which can be taken as future works. Second, the proposed method is only available for style conditions in the form of images. Other forms such as texts, videos and 3D data are not considered in this work and can be solved in the future.

**Broader impacts.**    Our work will not lead to significant negative social impacts. Problems such as privacy invasion and misinformation can be also attributed to normal image generative models. Solving such problems would be a large future research topic.

**Comparison with InstantStyle-SDXL and StyleAlign.**    In Fig. 9 we present the comparison between our method and InstantStyle and StyleAlign, both using SDXL as backbone network. We find that while InstantStyle does perform better with SDXL than SD1.5, suffering less from content leakage, it tends to generate images with classic art styles such as paintings. This makes the generated results less similar to the reference images. On the other hand, InstantStyle-SDXL still cannot handle styles such as cubism and impasto, which can be well modelled by our method. Moreover, StyleAlign is generally worse than the other two methods.

**Effectiveness of frequency domain decomposition.**    In the multi-source style embedding extraction module, we utilize the discrete wavelet transform to first decompose the patch level features of style reference images into low-frequency and high-frequency features. To see how this process can help our module, we visualize the attention weights for two sets of reference images among different denoising time steps in Fig. 10. Three main phenomenon can be concluded: (1) Style embeddings concentrate more on low frequency features, which is reasonable since low-frequency features contain information such as color. (2) The patterns of feature usage are consistent among two prompts for each style, while being different for different styles. (3) For high-frequency features, different timesteps generally focus on different information. These results can sufficiently support our design of decomposing the image features regarding frequency.

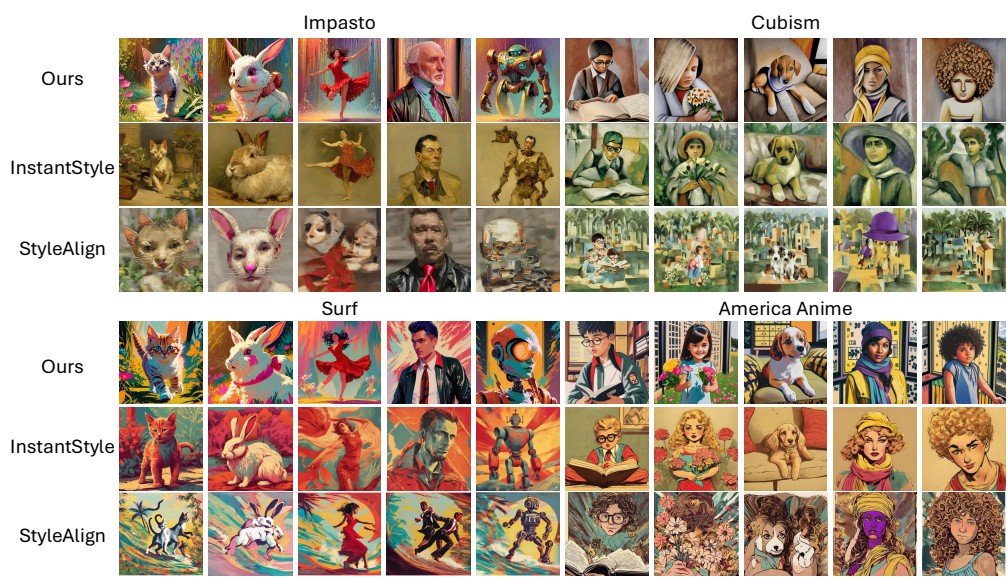

Figure 9: Qualitative comparison with InstantStyle and StyleAlign using SDXL as backbone.

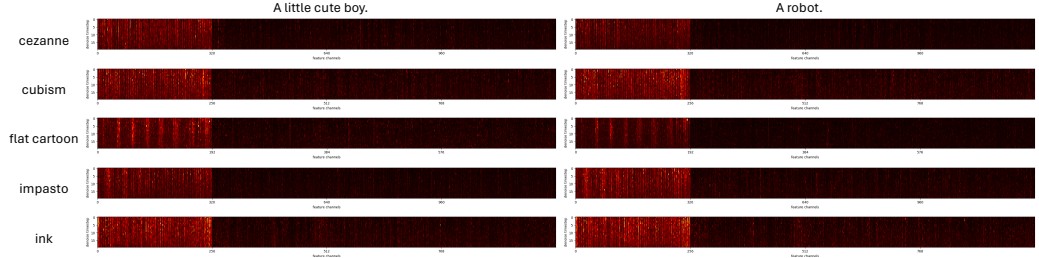

Figure 10: Attention weight visualization between style embedding and frequency domain features of style reference images. Each column represents the same text prompt and each row represents the same style, which are listed in the figure. Zoom in for more details.

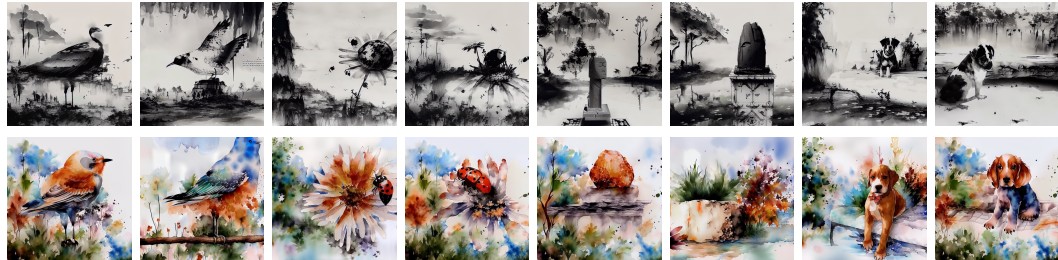

Figure 11: Multi-shot qualitative results with non-related objects added to the caption of style reference images. Prompts for every two columns from left to right: *A bird in a word; A daisy with a ladybug on it; A stone with a crack in it, holding a plant growing out of it; A puppy sitting on a sofa.*

**Versatility of our method.**    To show that our method is generalizable enough, we further apply our method to pretrained SDXL, which is an advanced version of SD. The results are shown in Fig. 13 and Fig. 14. We can find that basically the proposed method can introduce correct style to pretrained SDXL. Since there is significant gap between the prior knowledge learned by SDXL and SD1.5, the generated images also show different patterns. The results show that SDXL can better handle styles regarding lines and colors, while SD1.5 can provide better global-level styles. Moreover, SDXL can make better balance between style and general image aesthetic. The human faces generated by SDXL are more proper, thanks to its larger capacity.

**Reasonableness of negative semantic embedding.**    One would ask whether it is proper to directly subtract style token part in $\hat{z}_{caption}$ from $\hat{z}_{CLIP}$ and if the subtracted vector could inadvertently contain elements of the negative prompt (e.g., "reading a book"), rather than purely style information. Note that after each subtraction an attention is further applied to $\hat{z}^{CLIP}$, where unrelated negative prompts would be weakened. To verify this we provide several examples on multi-shot ink and watercolor styles. Specifically, we add a non-related caption '*There is a robot, a UFO and a monster in the image*.' to the caption of each reference image. The results are presented in Fig. 11, in which the semantics of generated images do not degrade compared with original StyleMaster.

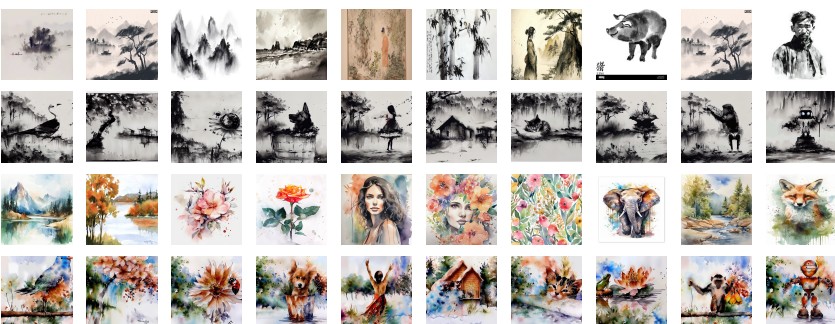

Figure 12: Qualitative results of StyleMaster under 10-shot setting. The first and third rows contain the reference images, the other rows show the generated results.

**Results with more shots.**    To further show the versatility of our proposed method, we conduct a 10-shot experiment, of which the results are presented in Fig. 12. The results show that our method, when given reasonable reference images, is robust and well performing under 10-shot setting.

**Image-to-image.**    Apart from the basic text-to-image task, we also extend our method to image-to-image task, in which we follow the commonly used pipeline to first and then adopt the ControlNet-Canny Zhang et al. (2023a) together with our proposed method to generate a new image with target style. The results are shown in Fig. 15. One can find that when given different reference images such as Japan anime, American anime, Von Goah's painting and pixar anime, etc. The human face

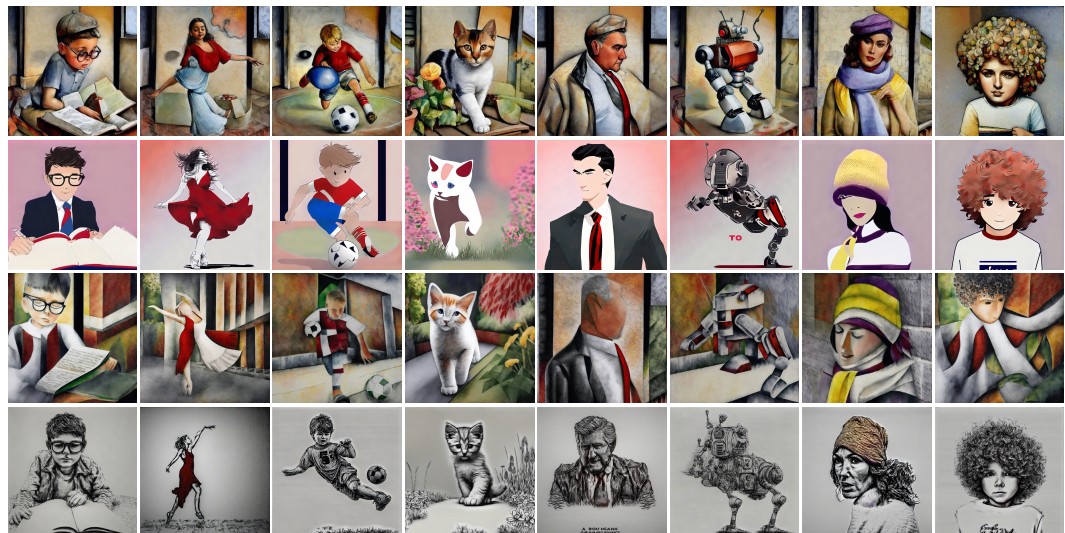

Figure 13: Quantitative results of our proposed method with SDXL in one-shot setting. Styles used in each row from up to bottom: *Cezanne, flat cartoon, cubism, pencil*.

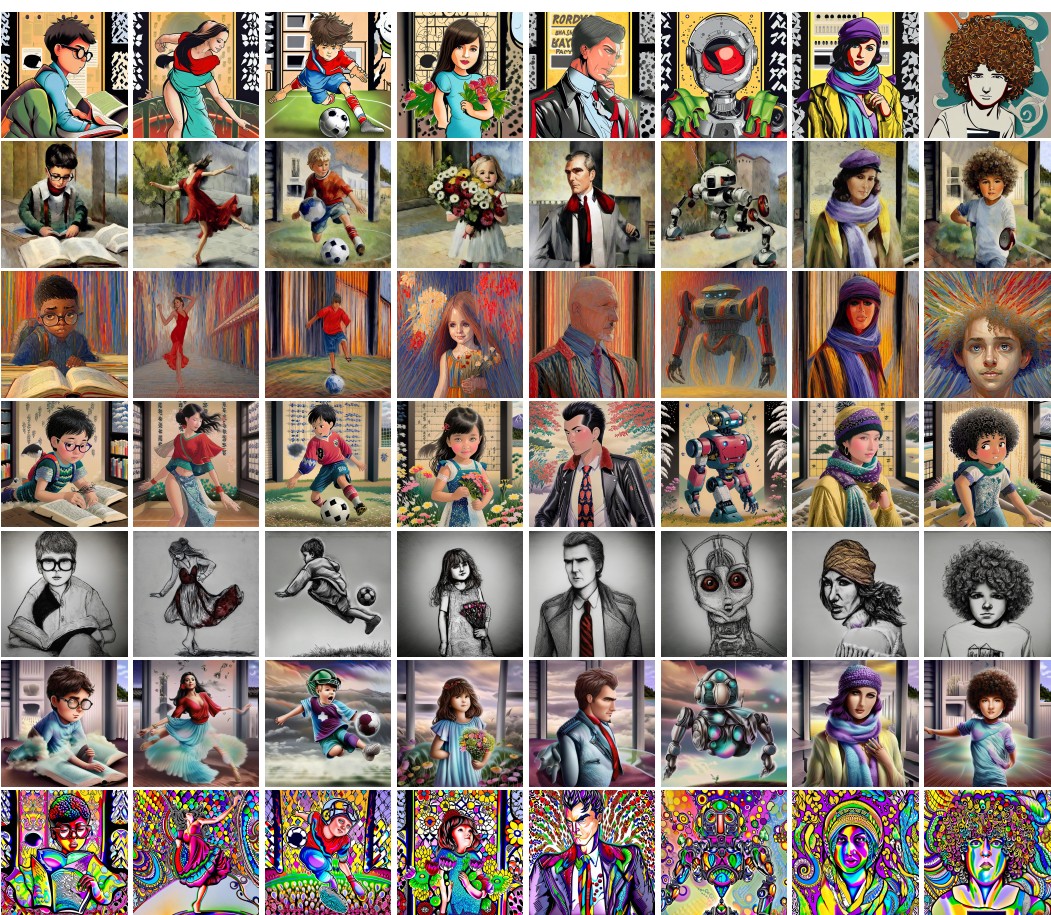

Figure 14: Quantitative results of our proposed method with SDXL in multi-shot setting. Styles used in each row from up to bottom: *america anime, Cezanne, Expressionism, ukiyoe, pencil, surreal, psychedelic*.

in the base image can flexibly change according to the style, which illustrates the effectiveness of our method.

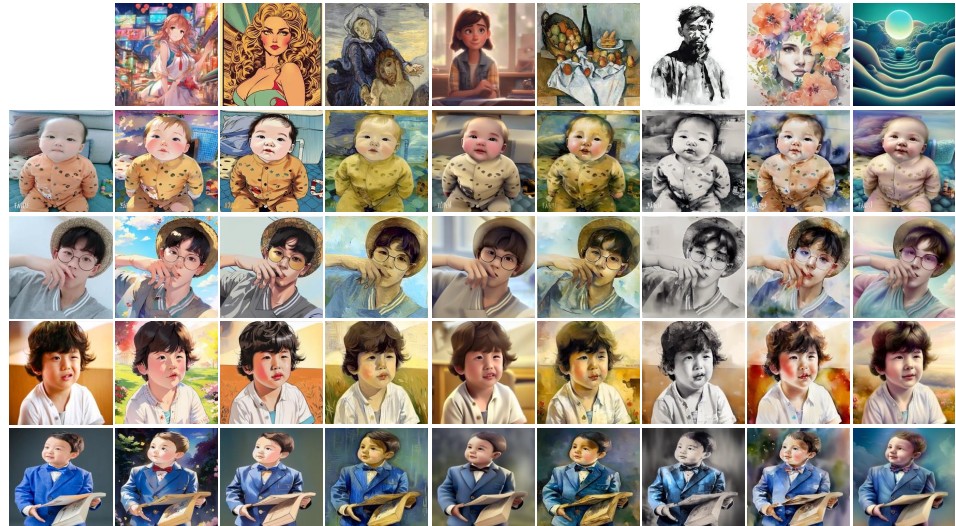

Figure 15: Image to image results generated by our model.

## C    MORE QUALITATIVE RESULTS

We provided more quantitative results in both one-shot and multi-shot settings in Fig. 16, Fig. 17 and Fig. 18. The images generated with each group of reference images share consistent style, while correctly showing the target objects.

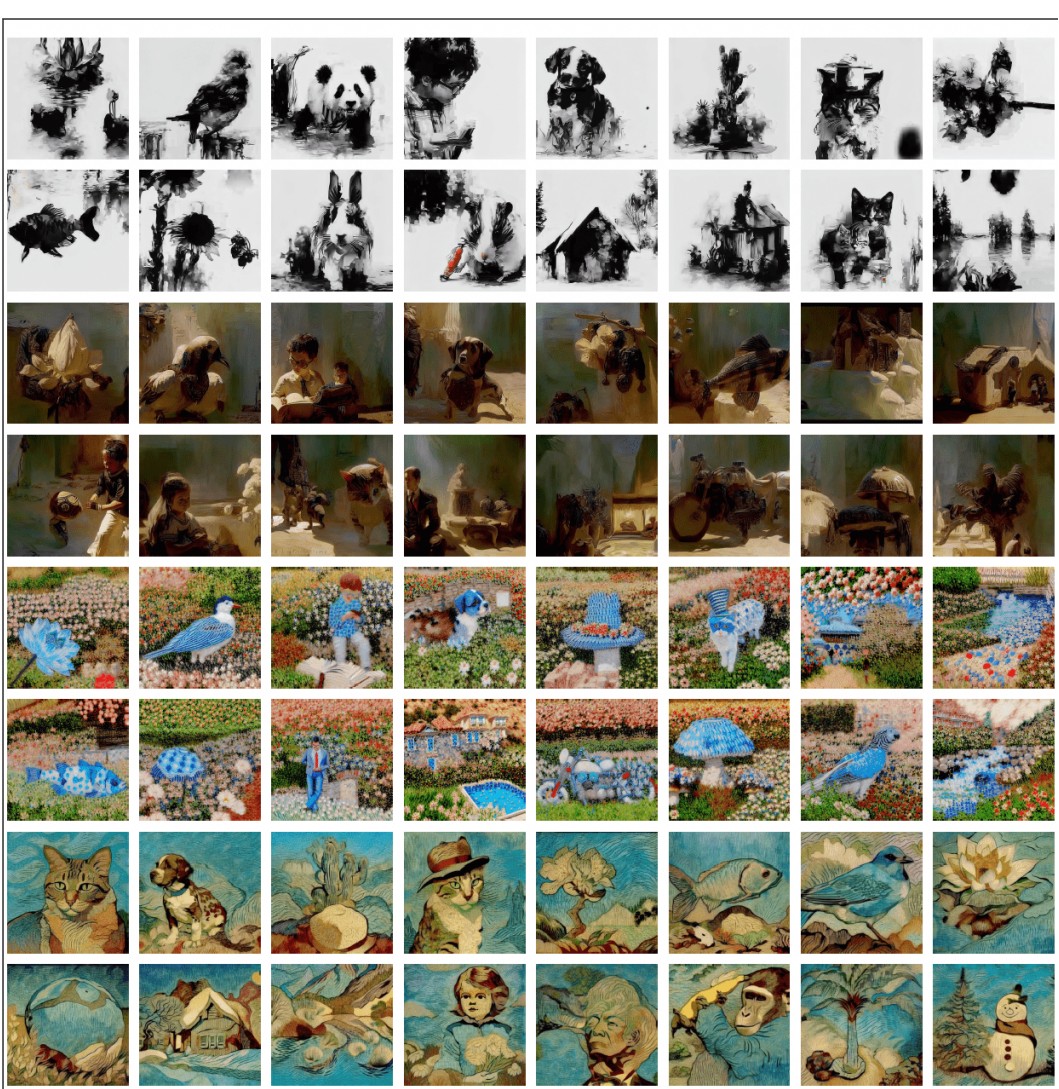

Figure 16: More quantitative results of our proposed method in one-shot setting. Styles used in each two rows from up to bottom: *ink, impasto, Monet, Van Goah*. Uncompressed version is in the supplementary material.

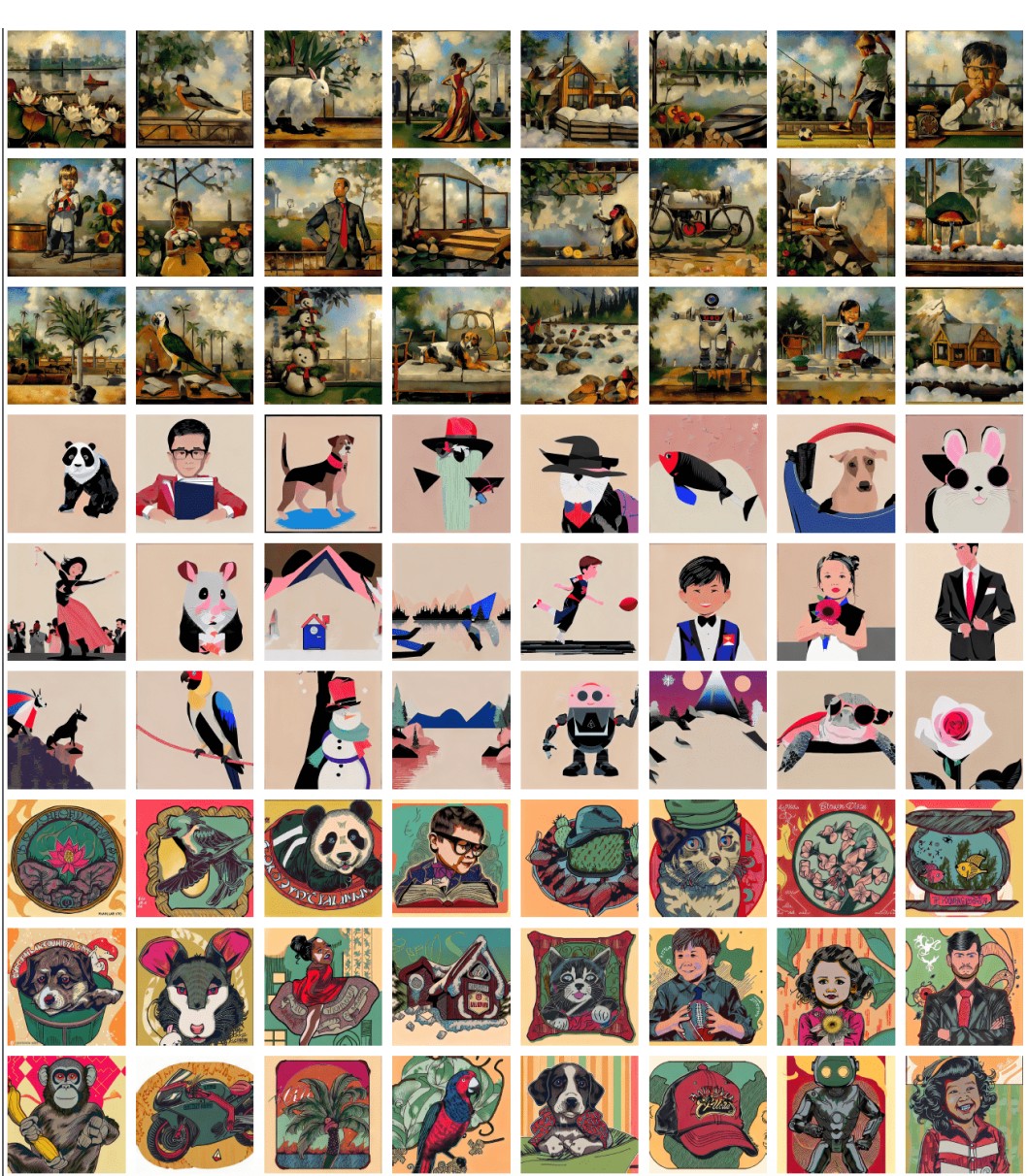

Figure 17: More quantitative results of our proposed method in multi-shot setting. Styles used in each three rows from up to bottom: *Cezanne, flat cartoon, america cartoon*. Uncompressed version is in the supplementary material.

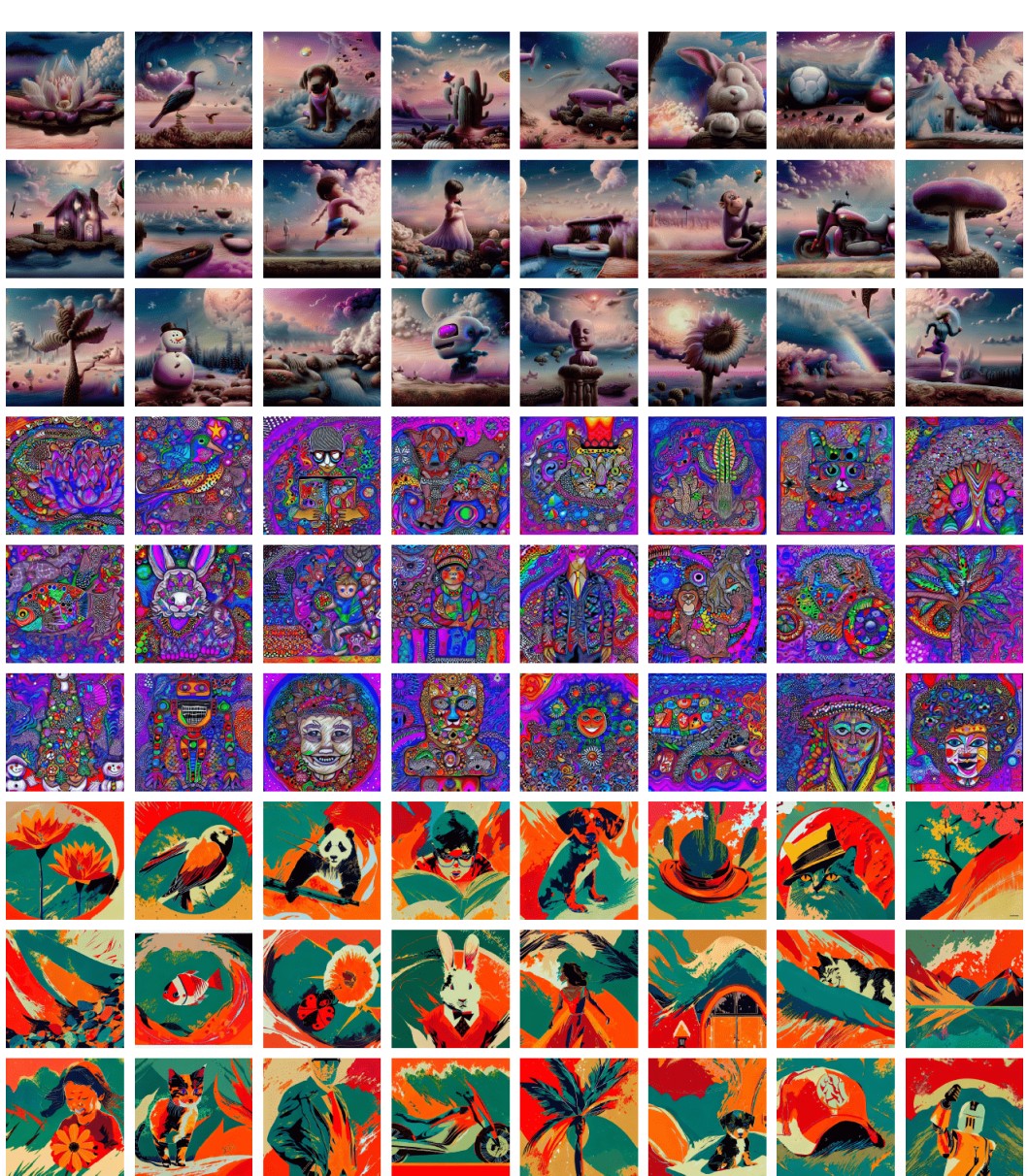

Figure 18: More quantitative results of our proposed method in multi-shot setting. Styles used in each three rows from up to bottom: *surreal, psychedelic, surf*. Uncompressed version is in the supplementary material.

