# OpenReview forum: "StyleMaster: Towards Flexible Stylized Image Generation with Diffusion Models"
_ICLR.cc/2025/Conference — ICLR 2025 Conference Withdrawn Submission_

### Official Review · Reviewer_qUmh · 2024-10-30

**Soundness:** 2
**Presentation:** 3
**Contribution:** 2
**Rating:** 5
**Confidence:** 4

**Summary:**

The paper studied stylized text-to-image generation and improved StyleAdapter by addressing the issues like misinterpreted style and inconsistent semantics. Specifically, the paper proposed 1) a multi-source scheme for style extraction, which incorporates global-level VGG descriptors and semantic text embeddings from image captions into style representations besides CLIP-based patch embedding; and 2) dynamic attention adapter layers to adjust the weights of the style embeddings to the self-attention and cross-attention layers in the diffusion UNet. The proposed method compares favorably with some recent SIG methods in the experiments.

**Strengths:**

This is a descent work on stylized text-to-image generation. Some technical approaches are interesting and may be inspiring to the community, like the dynamic cross attention adapter and multiple features to derive the style representations.

**Weaknesses:**

The major problem of this paper is that the claims are not well justified. There is some gap between the motivation of addressing the issues in StyleAdapter and the proposed technical approach.

Both CLIP features and VGG descriptors are trained or designed to conduct image/object recognition. So their abilities to describe and represent complicated or fine-grained image styles are in question, which lead to the questions: 1) If CLIP features are aligned with image captions and not good to describe image styles, VGG descriptors may also suffer from the similar issues; 2) Is image style more related to local image patterns or global content? More in-depth discussions on how to define and describe \emph{style} are required to justify the advantage of using VGG descriptors.

Conceptionally, if a method uses feature from more sources or removes some unrelated/negative tokens, it “may” improve the image generalization, but not necessarily. So the paper needs to explain and justify the proposed method can achieve this. For example, the paper needs to first justify “the global information can guide the model to better concentrate on style-related knowledge”, which is an assumption and shall not be used as a principle to derive the conclusion that so our approach can yield “better” results. Actually, the paper uses “better” many times where there are some logic gaps in those statements.

The embeddings of Z_{clip} and text embeddings Z_{cap} are no longer aligned after respective self attention operations. So it may not be that impactful to subtract/remove some unrelated/negative tokens as the paper claims.

The experiments show that the improvements are not consistent. There are 20 styles in the experiments, thus it is not clear how the proposed method performs for fine-grained style control.

**Questions:**

Please discuss “styles” are related to global image contents or local image characteristics.

Please explain more about why VGG descriptors that are trained for image recognition are good to describe image styles and do not suffer from the same issues as CLIP features.

Please discuss if the proposed can extend to handle hundreds of image styles.

Some missing related references, please compare and discuss:

Measuring Style Similarity in Diffusion Models, 2024.

StyleTokenizer: Defining Image Style by a Single Instance for Controlling Diffusion Models, ECCV 2024.

CLAP: Isolating Content from Style through Contrastive Learning with Augmented Prompts, ECCV, 2024.


Typo: ll.184, "are achieves".

---

### Official Review · Reviewer_JaeZ · 2024-11-01

**Soundness:** 3
**Presentation:** 2
**Contribution:** 3
**Rating:** 5
**Confidence:** 4

**Summary:**

This paper introduces StyleMaster to improve the style misinterpretation and semantic inconsistency in previous t2i generation frameworks. With proposed modules StyleMaster effectively combines style flexibility with text-driven semantic integrity.

**Strengths:**

StyleMaster shows improved style interpretation and semantic consistency compared to baseline models. I believe that this is valuable for users needing customizable style control in t2i generation.

**Weaknesses:**

1. The manuscript says, the use of the CLIP visual encoder for style embeddings emphasizes local patterns, but elaborating on this aspect could improve the manuscript. Specifically, (1) it’s unclear why local-only embeddings might be insufficient to replicate style, as I believe that local elements (e.g., brushstrokes) often receive primary focus from artists or viewers when evaluating specific artistic styles. (2) What if the CLIP visual encoder are used to model global features by embedding the entire image as a whole? I am not fully convinced of the need for an additional VGG network and Gram, though they are frequently used in style transfer. Is CLIP not sufficient for embedding global features?

2. The paper lacks a clear explanation and discussion on the necessity of semantic-aware features with BLIP in this model. Additionally, while this approach resembles the method in DreamStyler, the benefits or potential issues of omitting semantic-aware features remain unclear. Further discussion of how these features enhance overall performance, along with a comparison to DreamStyler, would clarify their contribution.

3. In model comparisons, there is a mix of models utilizing test-time optimization and those using pre-training approaches. Differentiating between these categories would improve clarity. Additionally, the N-shot experimental setup and procedure are somewhat ambiguous. I assume the initial (pre-trained) model is trained using the settings described in Section 5.1, but in the N-shot experiment, does further optimization occur using the N reference samples? If so, what approach is used?

**Questions:**

Please see the weaknesses.

---

### Official Review · Reviewer_75nS · 2024-11-02

**Soundness:** 2
**Presentation:** 3
**Contribution:** 2
**Rating:** 3
**Confidence:** 5

**Summary:**

The paper presents StyleMaster, a novel framework for Stylized Text-to-Image Generation (STIG), aiming to address issues in previous methods like misinterpreted style and inconsistent semantics. It proposes a multi-source style embedder to extract comprehensive style embeddings and a dynamic attention adapter for better style injection. The model is trained with additional objectives to enhance semantic and style consistency. Experiments show its superiority over existing methods in generating stylized images with correct styles and semantic fidelity.

**Strengths:**

1. The method is easy to follow. StyleMaster introduces a modular approach with each component focusing on a specific enhancement (e.g., style capture or semantic preservation), which effectively addresses common issues in stylized text-to-image generation.

2. The model consistently outperforms baselines in quantitative evaluations, including both objective measures (text and style similarity) and subjective user studies, showing its capability to produce high-quality stylized images.

**Weaknesses:**

1. The paper introduces several complex loss functions (e.g., Gram consistency loss and semantic disentanglement loss) and additional model components (e.g., multi-source embedding, dynamic adapter). While these designs are effective in improving performance metrics, they resemble "patches" added to the existing model, contributing to engineering complexity. The paper lacks an in-depth theoretical analysis to explain the interrelationships between these modules, relying instead on enhancing specific metrics with pre-existing components.
2. The multi-source style embedder and dynamic attention adapter largely extend existing techniques rather than introducing fundamentally new theoretical contributions. While effective in real-world applications, this approach may be perceived as lacking in novelty from a research perspective.
3. StyleMaster currently only supports image-based style conditioning. Extending this to other forms (e.g., text, video, or 3D data) would improve its versatility for broader stylization applications.
4. Due to the multi-source style embedding extraction, which involves patch-level transformers, the model is computationally intensive. This may limit its practicality in real-time or resource-constrained environments, particularly when handling larger or multiple style reference images.

**Questions:**

See Weakness

---

### Official Review · Reviewer_PMjk · 2024-11-02

**Soundness:** 3
**Presentation:** 2
**Contribution:** 3
**Rating:** 5
**Confidence:** 4

**Summary:**

This paper proposes to extract descriptive representations of style from both image and text, which are leveraged to steer the diffusion models via dynamic attention adaptation. Two additional losses (style disentagle loss and style consistency loss) are adopted to enhance the training.

**Strengths:**

1.Dynamic attention adaptation is exploited to generate additional weights of the attention layers in the diffusion model for each input style image.

2.Ablation study is sufficient.

**Weaknesses:**

1.The input style images should be put into Fig.1, 4, 5 and 6 for side-by-side comparsions.

2.The references of 2024 are missing in the part “Stylized image generation” of Sec.2. For example, [A-D] ...

[A] Style injection in diffusion: A training-free approach for adapting large-scale diffusion models for style transfer, CVPR'24

[B] ArtBank: Artistic Style Transfer with Pre-trained Diffusion Model and Implicit Style Prompt Bank, AAAI'24.

[C] DEADiff: An Efficient Stylization Diffusion Model with Disentangled Representations, CVPR'24

[D] ArtAdapter: Text-to-Image Style Transfer using Multi-Level Style Encoder and Explicit Adaptation, CVPR'24.

3.The details of style destroy are missing. How the input style image is processed in style destroy to derive the output?

4.The differences between results generated with or without style/disen loss are minor in Fig. 6 and Table.5.

**Questions:**

1.Why different styles are considered in one-shot (Fig.4) and multi-shot (Fig.5)? Does it mean that multi-shot inference cannot directly improve the one-shot one? Are there any trade-offs between one-shot and multi-shot settings?

2.Why the Text Sim of the run w/o disen loss (0.282) is higher than the run w/o style loss (0.280) in Table.5? In my opinion, the model trained w/o disen loss should be expected to entangle the style and the content, which leads to semantic leakage from the input style image to the generated image with a different prompt and thus lower Text Sim.

3.Will the training code be released in the future?

---

### Note · Authors · 2024-11-15

I have read and agree with the venue's withdrawal policy on behalf of myself and my co-authors.